# Contrast Enhanced Ultrasound Compared with MRI and CT in the Evaluation of Post-Renal Transplant Complications

Emanuele David [1,2,*], Giovanni Del Gaudio [3], Francesco Maria Drudi [3], Vincenzo Dolcetti [3], Patrizia Pacini [3], Antonio Granata [4], Renzo Pretagostini [5], Manuela Garofalo [5], Antonio Basile [6], Maria Irene Bellini [7], Vito D'Andrea [7], Mariano Scaglione [8,9,10,11], Richard Barr [12] and Vito Cantisani [3,*]

1 Department of Translational and Precision Medicine, Sapienza University of Rome, 00185 Rome, Italy
2 Unit of Radiology, Papardo Hospital, 98158 Messina, Italy
3 Department of Radiology, Policlinico Umberto I, Sapienza University of Rome, 00185 Rome, Italy; g.d.gaudio@gmail.com (G.D.G.); francescom.drudi@uniroma1.it (F.M.D.); vincenzo.dolcetti@hotmail.it (V.D.); patry.shepsut91@gmail.com (P.P.)
4 Nephrology Unit, Cannizzaro Hospital, 95026 Catania, Italy; antonio.granata4@tin.it
5 General Surgery and Organ Transplantation Unit, Department of Surgery, Sapienza University of Rome, 00185 Rome, Italy; renzo.pretagostini@uniroma1.it (R.P.); manuela.garofalo@uniroma1.it (M.G.)
6 Radiology Unit 1, Department of Medical Surgical Sciences and Advanced Technologies "GF Ingrassia", University Hospital "Policlinico-Vittorio Emanuele", University of Catania, 95123 Catania, Italy; basile.antonello73@gmail.com
7 Department of Surgical Sciences, Sapienza University of Rome, 00161 Rome, Italy; m.irene.bellini@gmail.com (M.I.B.); vito.dandrea@uniroma1.it (V.D.)
8 Department of Surgery, Medicine and Pharmacy, University of Sassari, 07100 Sassari, Italy; mariano.scaglione1@nhs.net
9 Department of Radiology, James Cook University Hospital, Middlesbrough TS4 3BW, UK
10 School of Health and Life Sciences, Teesside University, Tees Valley, Middlesbrough TS1 3BX, UK
11 Department of Radiology, Sunderland Royal Hospital, NHS, Sunderland SR4 7TP, UK
12 Department of Radiology, Northeastern Ohio Medical University, Youngstown, OH 44272, USA; rgbarr525@gmail.com
* Correspondence: emanuele.david@uniroma1.it (E.D.); vito.cantisani@uniroma1.it (V.C.)

**Abstract:** Renal transplantation (RT) is the treatment of choice for end-stage renal disease, significantly improving patients' survival and quality of life. However, approximately 3–23% of patients encounter post-operative complications, and radiology plays a major role for their early detection and treatment or follow-up planning. CT and MRI are excellent imaging modalities to evaluate renal transplant post-operative course; nevertheless, they are both associated with a high cost and low accessibility, as well as some contraindications, making them not feasible for all patients. In particular, gadolinium-based contrast can lead to the rare condition of nephrogenic systemic fibrosis, and iodine-based contrast can lead to contrast-induced nephropathy (CIN). CT also exposes the patients who may require multiple examinations to ionizing radiation. Therefore, considering the overall advantages and disadvantages, contrast-enhanced ultrasound (CEUS) is presently considered an effective first-line imaging modality for post-operative early and long-term follow-up in RT, reducing the need for biopsies and providing adequate guidance for drainage procedures. Hence, this paper aims to review the updated knowledge on CEUS compared with CT and MRI for the evaluation of RT renal transplant complications; advantages, limitations, and possible recommendations are provided.

**Keywords:** renal transplant; CEUS; US; Color-Doppler-US; complications

## 1. Introduction

Chronic kidney disease (CKD) is a condition in which the kidneys are diseased and have a decreased ability to filter blood. Excess fluid and waste in the blood remain in the body and may lead to other health problems such as heart disease and stroke.

CKD is defined when GFR is <60mL/min/1.73 m$^2$ for at least 3 months, in association with kidney parenchyma damage, assessed via markers as albuminuria, urine sediment, or a history of previous kidney transplantation. Approximately 37 million US adults are estimated to have CKD, with most remaining undiagnosed. Five stages of CKD are recognized, the last representing end stage renal disease (ESRD), requiring life-long renal replacement therapy (RRT) in the forms of dialysis or renal transplantation [1].

In end-stage kidney disease, kidney transplantation offers a survival advantage in comparison to renal dialysis, significantly improving patients' quality of life and outcomes.

According to the Organ Procurement and Transplantation Network (OPTN)/Scientific Registry of Transplant recipients (SRTR) Annual Data Report, in 2019 there were 24,273 renal transplants and more than 100,000 candidates on the waiting list. The mean patient's age is 50–60 years old. The average wait time to receive a renal transplant is at least 1 year [2].

Kidney transplantation (KT) is the treatment of choice for end-stage renal disease (ESRD), having the potential to reduce mortality and improve quality of life in comparison to chronic dialysis [3]. However, despite the developments in surgical techniques and immunosuppression regimens, post-transplant issues still represent a significant clinical problem in KT patients [4–7].

Complications can be preliminarily divided into early or late:

Early complications appear in the first weeks after transplantation and more often are related to surgical issues. Early complications include acute rejection, acute tubular necrosis, hematoma, pyelonephritis, abscess, urinoma, ureteral obstruction, vascular complications including arterial stenosis and thrombosis, arteriovenous fistula, arterial pseudoaneurysm, renal vein thrombosis, and graft torsion.

Late complications generally occur several weeks later and are usually due to medical problems usually related to immunosuppression and drug toxicity. Late complications include chronic rejection; other causes of ureteral obstruction, lymphocele, cyst, renal cell carcinoma, and transitional cell carcinoma of the graft; as well as complications due to immunosuppression such as lymphoma, Kaposi sarcoma, and opportunistic infections involving the graft [8–10] (Table 1).

**Table 1.** Most common parenchymal complications in renal transplantation.

- Immediate (within the first week)
  - Hyperacute rejection—accelerated rejection
  - Acute tubular necrosis
  - Acute tubular necrosis
  - Calcineurin inhibitors' toxicity
  - Infectious complications (acute graft pyelonephritis)
- Early (between the 1st and the 12th week)
  - Acute rejection
  - Calcineurin inhibitors' toxicity
  - Infectious complications (acute graft pyelonephritis)
- Late (after the twelfth week)
  - Chronic rejection
  - Calcineurin inhibitors toxicity
  - Infectious complications (acute graft pyelonephritis)
  - Nephropathy relapse

Complications can also be divided into nephrological, urological, vascular, and systemic causes. Nephrological complications include acute rejection, acute tubular necrosis, and cyclosporine toxicity; these are considered the most common causes of early graft failure following transplantation. Urological complications include urinary obstruction, of which the most common causes are strictures in distal third of the ureter, edema at the anastomotic site, blood clots within the ureter or bladder, and perinephric fluid collections (e.g., lymphocele, seroma, and hematoma). Stricture of the distal third of the ureter is

usually secondary to ischemia [11]. Urological complications also include renal calculus, urinary leak, lymphoceles, hematomas, abscesses, rupture, and torsion.

Vascular complications include transplant renal artery stenosis, transplant renal vein thrombosis, arteriovenous (AV) fistula, and pseudoaneurysms.

Ultrasonography (US) plays a major role in the imaging of these patients from the immediate post-operative period to long-term follow-up. Helical computed tomography (CT), magnetic resonance imaging (MRI), and angiography are used in indeterminate cases such as renal artery stenosis, thrombosis, and pseudoaneurysm. This paper reviews the updated knowledge on US, CT, MRI, and CEUS roles in this clinical setting, underlying advantages and disadvantages (Table 2).

**Table 2.** Indications for kidney transplant imaging.

- Routine surveillance imaging
- Immediate postoperative evaluation
- Fevers and chills
- Follow peritransplant collections
- Hypertension and/or unexplained graft dysfunction
- Elevated or rising creatinine
- Pain in region of transplant
- Severe hypertension refractory to medical therapy
- Decreased urine output

## 2. Imaging Modalities

### 2.1. Unenhanced US

US including color Doppler module has become the most suitable imaging tool to investigate the status of renal grafts in the first 24 hours after surgical transplantation. This technique plays a key role to investigate possible post-surgical complications including acute rejection and chronic allograft nephropathy [12]. In expert hands, color Doppler ultrasound can be a valid tool for the diagnosis and follow-up assessment of all vascular complications of renal transplantation [10,13–15]. Color Doppler US can demonstrate increased speed in transplant renal artery (TRA) stenosis, distal spectral lengthening, and increased arterial acceleration time of intra-parenchymal arterial vessels [16,17].

The cut-off for pathologic TRA peak systolic speed varies between 200 and 300 cm/s according to different authors. The lower value suffers from low specificity and can be responsible for an excessive number of superfluous investigations. Controversy still remains as to the best resistive index (RI) cut-off. CT angiography can be used to assess the exact location and degree of stenosis for possible subsequent interventional digital subtraction angiography (DSA). In this context, MRI angiography is a powerful alternative for detecting TRAS, although this imaging modality is less accessible, and may overestimate the degree of stenosis [18]. Nonetheless, given its high sensibility and specificity, CEUS can be used as a valid alterative to color Doppler to rapidly rule out TRAS, thus also splitting unnecessary CT angiography or DSA [19].

### 2.2. CT

CT represents a second-line imaging tool for assessing questionable US findings, offering an overall assessment of vascular complications, and identifying perirenal collections or renal neoplasms [20]. CT is less frequently used due to the need to administer iodinated contrast medium, which can be nephrotoxic. Contrast-enhanced CT can depict parenchymal, perirenal, renal sinus, pyeloureteral, and vascular diseases in renal transplantation in great detail despite the limitations.

Multiplanar and three-dimensional (3D) maximum-intensity-projection (MIP), shaded-surface-display (SSD), and volume-rendered reformatted images of the graft vessels and the recipient iliac arterial system and CT urograms are usually obtained in all cases.

All examinations are usually performed before and after injecting non-ionic contrast material to visualize arterial and renal venous phases, as well as, when pyelo-ureteral complications are suspected, a late phase complication [20].

### 2.3. Magnetic Resonance Imaging

MRI has several advantages, such as the lack of ionizing radiation and the possibility to obtain relevant tissue information without the addition of a contrast agent, reducing the risk of contrast-induced nephrotoxicity. When contrast enhancement is needed, the risk of gadolinium-related safety issues used in patients with impaired renal function has to be considered [11]. However, limited availability and other limitations such as high cost, long examination time, and the lack of portability make it less attractive. Additionally, it provides no option for direct intervention which, for example, digital subtraction angiography (DSA) can offer. Despite promising outcomes, MRI is still infrequently used in the post-transplantation diagnostic process [21].

### 2.4. CEUS

Contrast-enhanced ultrasound (CEUS) is playing an increasing role in the assessment of KT complications. CEUS permits the assessment of both macro- and micro-vascularity, as well as perirenal fluid and all the parenchymal abnormalities related to rejection, including acute tubular necrosis (ATN), vascular complications, and parenchymal tumors [22]. CEUS also allows the detection of other vascular complications including thrombosis of the renal vein and anatomical variants such as a hypertrophied column of Bertin (Figure 1).

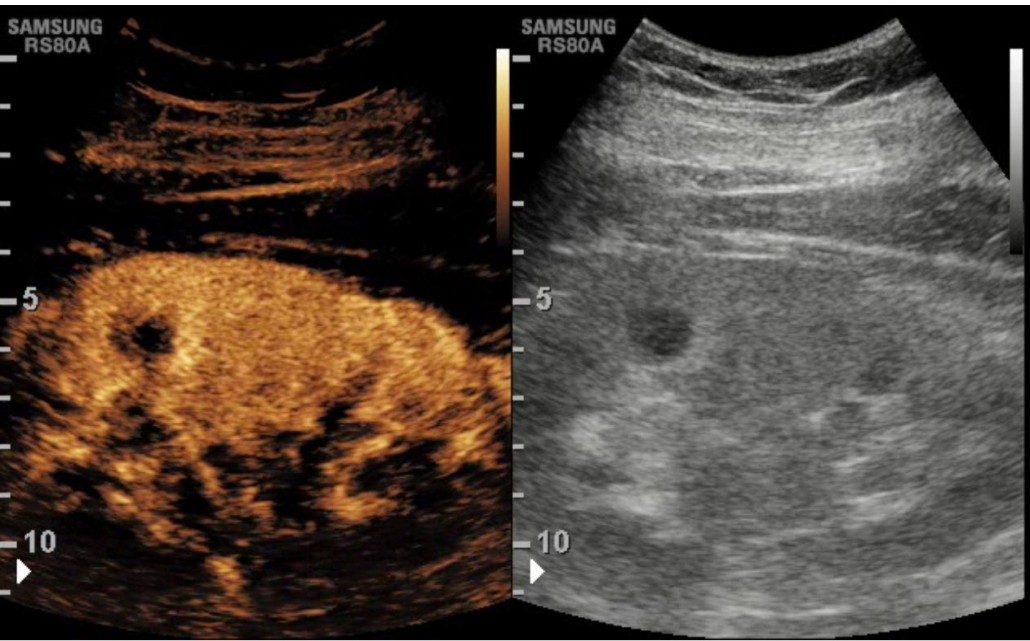

**Figure 1.** At CEUS, the hypertrophic pseudonodular appearance of the middle renal cortex shows the same appearance of the remaining cortex, thus confirming it is a hypertrophied column of Bertin.

CEUS allows a real-time assessment of contrast enhancement, with the ability to record the entire real-time, high frame rate examination as a cine-loop for review. Real-time acquisition allows for the representation of the cortico-medullary phase in all examinations, independently from the patient's hemodynamic status, without the need for bolus tracking. Continuous data acquisition allows for time-intensity curves evaluating contrast enhancement intensity versus time, from which extracting quantitative indexes of perfusion is possible with the appropriate software [23].

CEUS is safe without any radiation exposure; it is repeatable and avoids potential nephrotoxic effects of iodinate contrast agents. All the above-mentioned features aid in

both the postoperative recovery of renal function, as well as in the follow-up of chronic graft impairment cases [24].

Therefore, CEUS can be the suggested first-line examination to assess post-KT related complications, thus selecting patients for prompt intervention, surgery, or follow-up. However, it should be underlined that CEUS requires sufficient skill in the examination technique, and, above all, is less panoramic compared to CT and MRI [24,25].

Furthermore, to date, there is no evidence showing significant superiority of CEUS over color-Doppler Ultrasound (CDUS) in the diagnosis of TRAS. CEUS, however, because of its repeatability, low cost, ease of performance even at the patient's bedside, and absence of nephrotoxicity, has significant diagnostic potential, especially in cases where, as reported above, Doppler methods have limitations.

A comparison of CEUS vs. US Doppler, CT, and MRI, comparing their efficacy in all the major issues of post-transplant kidney evaluation, limitations, and future directions, is presented in Table 3.

**Table 3.** Comparison of Pros and Cons for CEUS, US Doppler, CT, and MRI.

| | PROS | CONS |
|---|---|---|
| CEUS | Lack of ionizing radiation<br>Inexpensive<br>Repeatable<br>Real time examination<br>Fast<br>Macro and micro vascularization assessment<br>Safe in patients with renal impairment<br>Can be performed at the bedside (no need to transport the patient)<br>Can be used to guide procedures | Absence of wide view compared to CT and MRI.<br>Requires experienced operators. |
| US with Doppler | Follows most of the advantages of CEUS;<br>It doesn't use contrast media | Requires an expert operator;<br>Affected by artifacts |
| CT | Panoramic view<br>High spatial resolution<br>Volumetric rendering<br>Fast | Nephrotoxic contrast medium<br>Ionizing radiation<br>Not feasible in patients with high creatinine blood values |
| MRI | Panoramic view<br>Lack of ionizing radiation<br>Tissue characterization | Nephrotoxic contrast medium<br>Expensive<br>Time consuming |

*2.5. CEUS Technique*

The current standard for CEUS, which uses the same transducer used for abdomen imaging, uses second-generation ultrasound contrast agents (UCAs), consisting of a phospholipidic or albumin shell containing microbubbles of an inert lipophilic gas, such as sulfur hexafluoride [SonoVue©/Lumason© (Bracco, Milan, Italy)]. The contrast agent is injected intravenously and based on blood-pool pharmacokinetics; it remain in the intravascular compartment for some time before dissolving through the lungs (gas component) and being metabolized by the liver (compatible shell) [16]. After B-mode and color Doppler evaluation, CEUS is performed with an intravenous bolus administration through a ≥20 Gauge catheter. In general, a single dose of less than 2 mL of Sonovue© is sufficient for KT-related applications, followed by a 5–10 mL saline flush. When using Definity, 0.5 mL is injected followed by the saline flush. Care must be taken on how the contrast is injected to prevent busting of the bubbles [26]. CEUS needs a specific ultrasound contrast imaging mode to prevent microbubbles from being destroyed by elevated acoustic power, e.g., use of a low mechanical index [16]. In addition to the low MI technique, contrast can be injected to salvage a limited Doppler examination by enhancing the Doppler signal. However, with the high MI, the enhancement is short lived so the low MI technique is favored [27].

The following post-contrast phases can be observed:

- A cortico-medullary phase is an arterial phase in which the renal cortex is mostly enhanced.
- A nephrographic phase provides a more homogeneous enhancement of the cortex and medulla, at 30–70 s after injection.
- The delayed imaging achieved after >70 s from contrast injection does not represent a urographic phase because such contrast agents in US imaging are not actually excreted by the kidney [15].

CEUS examinations are documented by acquiring cine loops, which are prospectively or retrospectively stored in a picture archiving and communication system (PACS) [19].

### 3. CEUS Role for Transplant Kidney Complications

Perirenal collections are the most common post-surgical complications after KT. They are frequently asymptomatic, although large ones may cause displacement of the graft or cause compression on the vessel or ureter [28]. US has a limited value in detecting symptomatic collections and tends to misjudge their size. Both CT or MRI provide a more reliable representation of such collections and can better depict its content. The use of UCAs enhances the visualization of vessels, graft parenchyma, adjacent tissues, and collections. According to Grzelak et al., the injection of CEUS permitted the detection of 17.6% more cases of perirenal hematomas compared to US alone, allowing the identification of collections with a wall thickness lower than 10 mm [15,16]. Graft Rejection, although decreasing in frequency in recent years, remains one of the main complications after KT, with an incidence of 9% [29]. With US, rejection has a loss of cortico-medullary distinction and increased cortical thickening. Doppler US can show an increase of RI, although this feature is not typical [12,21]. A major advantage of CEUS in this setting is a perfusion analysis using various quantitative values, a finding that showed a potential risk for rejection [30]. Benozzi et al. demonstrated an increased time-to-peak in patients with acute rejection compared to a control group. In this study, patients with acute tubular necrosis showed a wide range of anomalies including lower cortico-medullary ratio of mean transit time and regional blood volume, suggesting that perfusion assessment might be helpful in distinguishing rejection from other causes of early graft impairment [31].

Transplant renal artery stenosis (TRAS) is a vascular complication that generally occurs between the 3rd and 24th month after kidney transplantation. It has an incidence and prevalence ranging from 1 to 23% and 1.5 to 4%, respectively [25–27]. If missed, stenosis can progress to renal transplantation loss. Clinically, because of its nonspecific clinical manifestations, it can be difficult to detect. Therefore, early imaging diagnosis of TRAS is of paramount importance because it is a potentially reversible condition whose treatment can lead to recovery of renal function and blood pressure control. Color Doppler ultrasound (CDUS) is the test of first choice in the diagnosis of TRAS [28]. Doppler criteria for TRAS includes: PSV > 200 cm/s, a velocity gradient between stenotic and pre-stenotic (iliac vessel) segments of 2:1 [32]. Its limitations are due to the marked tortuosity of transplant vessels that can lead to erroneous insonation angles, reducing the accuracy of Peak Systolic Velocity (PSV) measurements. Other sources of error are renal artery stretching and/or kinking, which can lead to a high peak systolic velocity and thus a false-positive diagnosis of TRAS [10]. In our experience, the diagnosis of TRAS can be made in most cases with CDUS. In daily clinical practice, the CDUS diagnosis of TRAS may run misdiagnosed in cases where TRAS is secondary not to geometric stenosis but to functional stenosis related to graft position. In these cases, CEUS has been shown to be able to detect changes in renal perfusion resulting from stenosis [9]. In fact, using time-intensity curves, a quantitative assessment of renal perfusion can be obtained [10]. Parameters such as time to contrast medium inflow into the renal cortex, duration of time to peak, peak index, ascending slope of the curve, and intensity curve allow precise quantification of renal perfusion that correlates with renal function [12]. A significant prolongation of both contrast-enhanced inflow to the cortex and the renal pyramids was reported by Rennert et al. [33] in patients with transplant renal

artery stenosis compared with normal individuals. Indeed, CEUS enables to identify the narrowed arterial segments and by means of TIC curves, prolonged inflow time is identified. When US methods are inconclusive and vascular pathology of the transplanted kidney is suspected, CT angiography or MRI are the imaging modalities of choice. Thrombosis of the transplanted renal vein (TTRV) is also a rare event: it usually occurs early after surgery with a reported prevalence of 0.1% to 4.2% [34]. It can be complete, in which case it involves loss of the graft, or partial loss. CDUS plays an important role in the diagnosis and follow-up of this complication. In the presence of complete TTRV, the following are observed: kidney enlargement, reduced parenchymal echogenicity, reduced/absent corticomedullary differentiation, the disappearance of renal sinus and collector system (all nonspecific features), and poor or absent compressibility of the vessel. The two most important findings of CDUS are the absence of venous signal venous signal (reflecting the absence of vasculature) and the diastolic reverse flow in the renal artery. There are no studies reporting the role of CEUS in the diagnosis of TTRV and CE-TC, showing that delayed nephrogram is the method of choice. TTRV stenosis may be secondary to extrinsic compression of a perinephric fluid collection (e.g., lymphocele, etc.) or perivascular fibrosis. At CDUS, renal parenchyma may appear normal or mildly hypoechogenic and essentially nonspecific, whereas a 3–4-fold increase in PSV between stenotic and pre-stenotic segments is considered highly suggestive of focal stenosis. Again, there are no studies demonstrating the role of CEUS in this as opposed to the heavy methods that are of choice. Angiography should be used only in doubtful cases to confirm and/or treat the stenosis. The transplanted kidney can be placed peritoneally or extraperitoneally, and in both cases, although rarely, complications represented by torsion and compartmental syndrome of the transplanted kidney, respectively, can occur. Torsion of transplant kidney can occur in the first week, but also several months after transplantation. Again, early diagnosis is critical to ensure transplant survival. Unfortunately, most patients are asymptomatic, and therefore the diagnosis can be made either during follow-up or in individuals with unexplained decline in renal function. It should be suspected in the presence of a change in renal axis from baseline. The extent of vascular flow impairment depends on the degree and duration of torsion and can result in extremely varied arterial flow. Although poorly specific, other findings include changes in parenchymal echogenicity, hydronephrosis if the ureter is not kinked, and urothelial thickening. CEUS does not appear to provide useful elements for diagnosis. The presence of surgical wounds or dressings can alter the acoustic window and cause a spurious variation in the renal axis. Images in the transverse plane are less operator-dependent and should be used as a baseline. As a result of the poor diagnostic accuracy of US imaging, computed tomography (CT) is useful to confirm the diagnosis. Compartment syndrome (CS) is observed in approximately 1–2% of cases and is an under-recognized cause of early dysfunction or loss of the transplanted kidney. When the kidney is placed extraperitoneally, i.e., between the anterior abdominal wall and the parietal peritoneum, it can be subjected to significant compression, which in the immediate postoperative period can result in hypoperfusion of the parenchyma. Therefore, extraperitoneal placement in the iliac fossa potentially predisposes the graft to secondary CS ischemia. The risk of CS correlates significantly with graft volume and usually occurs within the first 2 h after transplantation. If recognized early, related decompression surgery can result in the recovery of kidney function. B-mode US is generally nonspecific, whereas power and b-flow methods that are highly sensitive to the slow flows typically found intraparenchymally may show cortical hypoperfusion. CD-US is of little use as a result of low sensitivity to slow flows, but spectral analysis can show typical parvus-tardus in the arcuate arteries in presenting main renal artery pervia. Tardus parvus refers to a pattern of Doppler ultrasound spectral wave form resulting from arterial stenosis. The phenomenon is observed downstream to the site of stenosis and is due to reduced magnitude of blood flow through the narrowed vessel during ventricular systole. This characteristic pattern is useful in assessing renal artery stenosis. The pattern can also be seen distal to other sites of arterial stenosis (e.g., hepatic arterial stenosis), as well as within collateral vessels following

an arterial occlusion [35]. The VPS observed in the main renal artery may be very high in the presence of kinking or very low in the absence of kinking. The finding of an isolated elevated VPS of the renal artery with normal CD-US immediately after transplantation should not be attributed to CS but, instead, to perivascular edema and/or the patient's hemodynamic status. The literature does not report the role of CEUS in CS, but the experts' experience is that compared with power or b-flow it does not seem to add any additional information. For the study of slow flows, such as intraparenchymal flows, CDUS has little diagnostic value. Transplant renal artery stenosis (TRAS) following kidney transplantation is a possible cause of graft failure. It is considered the most common vascular complication (75%) in renal. The most frequent localization of stenosis is para-anastomotic (ranging from 25% to 78%). Patients can be treated by percutaneous transluminal angioplasty (PTA) or percutaneous transluminal stenting (PTS). The 12 month patency rate after IR ranges from 72 to 94%. The overall complication rate is 9%, with pseudoaneurysms and hematomas as the most frequent complications. TRAS can be successfully and safely treated through an endovascular approach. Stent delivery seems to guarantee a higher patency rate compared to simple angioplasty; however, further studies are needed to confirm these results [36]. Transplant renal artery and vein thrombosis have an early onset and a dramatic clinical manifestation and usually lead to allograft loss. Transplant renal vein thrombosis (TRVT) is dramatic and remains one of the most important causes of graft loss during the first month post transplant [37]. Living kidney donation (LKD) offers the best treatment for people suffering from end stage renal disease (ESRD). Worldwide, most transplanted kidneys come from deceased donors, but research shows that LKD offers numerous advantages to the recipient, namely pre-emptive transplantation, with the possibility to plan the surgery in advance and less immunological and ischemic insults, since the retrieved organ often comes from the same hospital and/or a well-matched donor [38]. Right-sided living donor kidneys have longer renal arteries and shorter veins that make vascular anastomosis more challenging, and technical challenge is the most common cause of early graft loss. The risk of early graft loss among recipients who received right kidneys is doubled compared to those who received left living donor kidneys, in fact, transplant surgeons prefer left-sided living donor kidneys because the longer renal vein facilitates implantation of the donor kidney to the deeply situated recipient right iliac vein [39]. There are also systematic complications such as neoplasms and or infection [8–10].

## 4. Discussion

KT remains the most effective method of treatment for ESRD, reducing the mortality rate and improving the quality of life compared to the alternative of chronic dialysis [4]. However, despite the advances in surgical techniques and immunosuppression therapy, significant post-transplant complications in KT recipients may occur [5]. CEUS represents a powerful tool to improve the speed and cost-effectiveness of the diagnostic work-up after KT. US is portable and can be used at the patient's bedside, avoiding transporting critically ill patients from the intensive care unit to the radiology department. The transplanted kidney examination in the iliac fossa can be performed rapidly and without the need to control respiration. This is an extraordinary benefit compared to more time-consuming imaging techniques, especially MRI. CEUS uses a non-renal toxic contrast media based on microbubbles of an inert gas, i.e., sulfur hexafluoride or perfluorocarbons, which are shattered by the high-energy ultrasound waves, thus providing information about flow and tissue perfusion. Compared to contrasts used in CT or MRI, they are pure intervascular agents that do not extravasate into the vascular system. They are non-nephrotoxic and can be safely tolerated in patients with renal impairment or renal occlusion [40]. Contrary to CE CT or MRI, in which the acquisition is performed at definite time-points (i.e., snap shots in time), CEUS provides a real-time, high frame rate representation of contrast distribution, with the ability to record the entire contrast examination with dynamic cine-loops. Real-time acquisition allows a correct representation of the cortico-medullary phase in all examinations, independently from the patient's hemodynamic status, without the need for

bolus tracking. Continuous data acquisition can be also translated into time-intensity curves (TIC), plotting contrast enhancement intensity versus time, which in turn are the bases for extracting quantitative indexes of perfusion with the proper software [22] (Figures 2 and 3). To summarize: CEUS represents an ideal first-line tool in evaluating KT patients, improving the accuracy of conventional US for a variety of complications. However, it is not free from limitations: CEUS requires skilled operators and is also less panoramic than CT and MRI. Additionally, not all patients are eligible such as uncooperative patients. CT may be considered a complimentary tool in the evaluation of RT complications. In patients with a satisfactory renal function, contrast-enhanced CT angiography (CTA) can be used to assess and treat a renal artery stenosis. Furthermore, in the delayed acquisition, contrast-enhanced CT can be used to confirm or exclude a urinary leakage or urinoma. Non-contrast CT may be helpful to assess the extent of a perinephric fluid collection and its relationship to the adjacent structures. Finally, unenhanced CT performs a better assessment of renal and ureteral stones than US, especially for small stones [41,42].

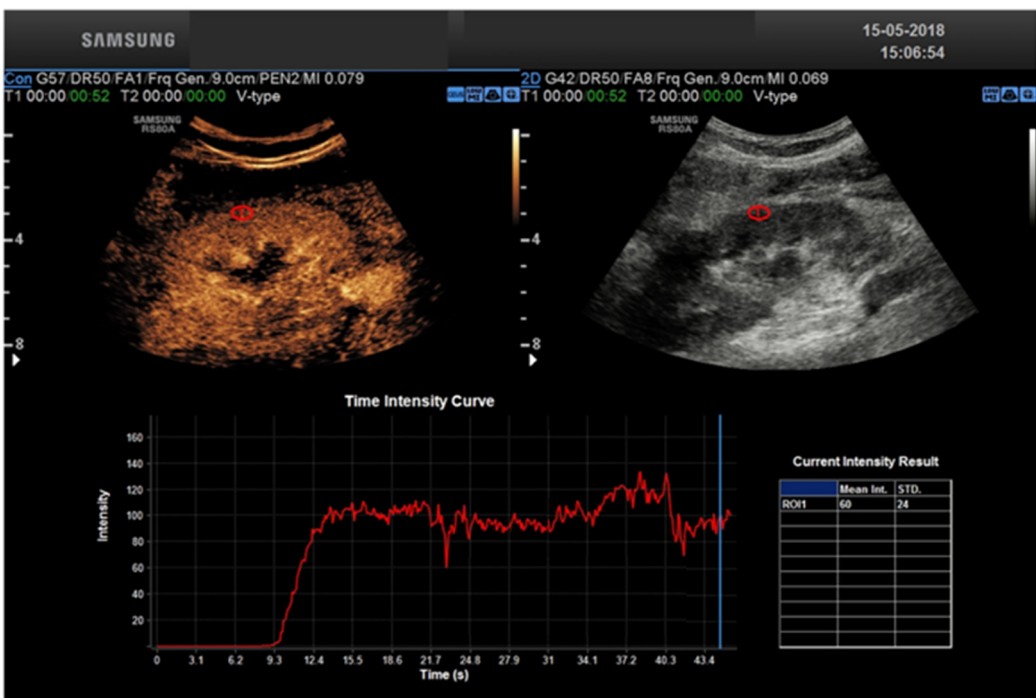

**Figure 2.** Healthy kidney TIC.

MR imaging (MRI) has gained an ancillary role in the evaluation of renal transplant complications. Magnetic resonance angiography (MRA) can be used to noninvasively evaluate for renal artery stenosis. Contrast-enhanced MRI can also provide a complete characterization of focal renal lesions, depicting their exact components as cystic or solid. Moreover, similar to CT, excretory phase imaging can be performed to assess a urinary leakage or urinoma. The injection of gadolinium-based contrast agents in individuals with impaired renal function has been referred to the possible development of nephrogenic systemic fibrosis, which is a potentially lethal occurrence.

However, gadolinium-based contrast agents used today are associated with few, if any, uncompounded cases of nephrogenic systemic fibrosis (NSF). Further understanding of their potential role in the evaluation of individuals with impaired renal function is under consideration [37,38].

The use of non-contrast-enhanced MRA (NCE-MRA) techniques is being encouraged in patients with an increased risk of NSF due to their renal insufficiency.

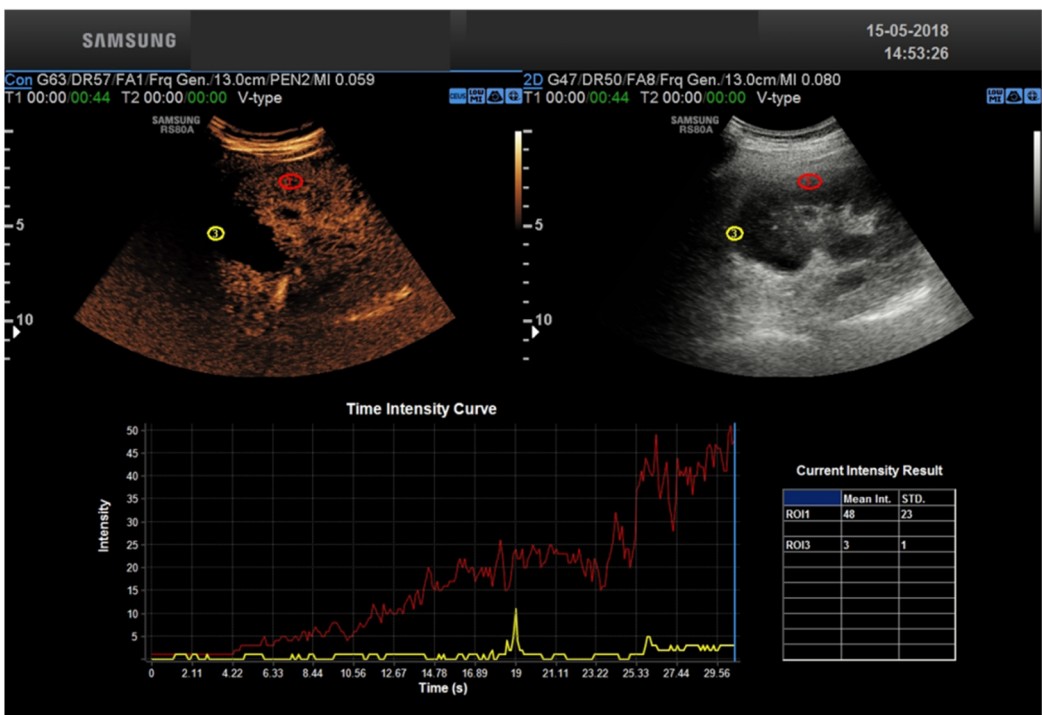

**Figure 3.** Ischemic Kidney TIC.

Ali Serhal et al., showed that NCE-MRA is a risk-free alternative to CTA and contrast-enhanced MRA (CE-MRA) in the evaluation of arterial anastomoses in renal transplant patients. However, it needs further confirmation [43–45].

In conclusion, following ALARA (As Low As Reasonably Achievable) criteria and the above-mentioned advantages, according to our experience and existing literature knowledge, CEUS is an effective imaging technique to evaluate post-transplant kidney disease and should be included in the prompt work-up of these patients.

**Author Contributions:** Conceptualization, E.D. and G.D.G.; methodology, R.B.; validation, F.M.D. and M.G.; formal analysis, V.D. (Vincenzo Dolcetti); data curation, M.I.B. and A.G.; writing—original draft preparation, M.S.; writing—review and editing, A.B. and M.S.; visualization, P.P.; supervision, V.D. (Vito D'Andrea) and R.P.; project administration, V.C. All authors have read and agreed to the published version of the manuscript.

**Funding:** This research received no external funding.

**Institutional Review Board Statement:** Not applicable.

**Informed Consent Statement:** Not applicable.

**Data Availability Statement:** The data that support the findings of this study are available at National Center for Biotechnology Information (NCBI) [Internet]. Bethesda (MD): National Library of Medicine (US), National Center for Biotechnology Information; [1988]. Available from: https://www.ncbi.nlm.nih.gov/, accessed on 17 May 2022.

**Acknowledgments:** We would like to thank Denise V. Nemeth, MPAS, PA-C from the University of the Incarnate Word in San Antonio, TX, and Karen Outtrup for the help and support in the language editing of this manuscript.

**Conflicts of Interest:** The authors declare no conflict of interest.

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
