# Peer review of "Contrast Enhanced Ultrasound Compared with MRI and CT in the Evaluation of Post-Renal Transplant Complications"

_tomography, doi:10.3390/tomography8040143_

Round 1

Reviewer 1 Report

This paper does not add to the readers knowledge as does not perform a systematic review or meta analysis of the relevant literature. The introduction does not cite the current incidence of renal artery stenosis, need for intervention, venous stenosis, graft technical loss rate, differences between right grafts and left grafts or live donor vs deceased donor grafts. It essentially just says that CEUS is a safe modality and useful, without any obvious evidence to compare with standard Doppler USS. Little referencing idnetifying the incidence of Tx vascular complications is mentioned and no comparison as to whether CEUS picks up a higher incidence of renal artery stenosis than Doppler US or CT angio. Can it identify torsion , can it identify compartment compression?? The suggestion that this technology can diagnose rejection is also misleading.  

Author Response

Dear Reviewer, thank you so much for giving us the opportunity to improve our paper.
We have edited the paper based on suggestions made by both reviewers; we used track changes for typos corrections in the manuscript and underlined in yellow the added paragraphs.
More in the details, following Reviewer 1 suggestions, we cited the current incidence of renal artery stenosis, need for intervention, venous stenosis, graft technical loss rate, differences between right grafts and left grafts or live donor vs deceased donor grafts adn reported more extensively the updated knowledge on Imaging role for the evaluation of the different complications.
We also expanded the references about the incidence of Tx vascular complications by comparing CEUS, Doppler and CT and we have dealt with torsion and compartment compression in dedicated sections and in the details.

Reviewer 2 Report

The authors present a review paper on the Contrast Enhanced Ultrasound Compared with MRI and CT in the Evaluation of Post-Renal Transplant Complications.

The topic is important and relevant to the scope of the journal .

There are several discrepancies, that must be solved:

Multiple typing errors (e.g. line 118, 137, 159-160, etc.). 

Omniscan, OptiMARK and similar linear structure MRI contrast agents are suspended in the EU market since 2017, so it‘s not fair to include them as a „disadvantage of MRI“.

If the authors speak about stenosis of renal arteries - what about new techniques of non-contrast MRA (NCE-MRA) for tranplant kidney arteries? The new techniques should be addressed in the article. 

Sincerely, 

Author Response

Dear Reviewer, thank you so much for giving us the opportunity to improve our paper.
We have edited the paper based on suggestions made by both reviewers; we used track changes for typos corrections in the manuscript and underlined in yellow the added paragraphs.

We have corrected the typing errors reported, we have eliminated the part concerning contrast media which are no longer in use, such as Omniscan and OptiMARK; furthermore, using the current literature we have dealt with the topic concerning new techniques of non-contrast MRA (NCE-MRA).

Round 2

Reviewer 1 Report

This is now much better than the initial version, but having added more useful information there is now some repetition of information regarding the incidence of TRAS and RV thrombosis. Apart from the repetition, the quoted incidence of over 4% for renal vein thrombosis I presume is historical as many have not seen this complication for over a decade. Worth mentioning the range and quoting some registry data for kidney allograft loss at 1 year which for some countries for all causes is less than 4%. Overall this is much improved because of it's clinical relevance, but as this is not a systematic review without any assessment of quality of data or meta analysis  it does not convince the reader of any new information. It is a useful revision of vascular complications however. Worth explaining what parvus tardus means for the readers .

Author Response

Dear Editor and Reviewer, thank you so much for giving us the opportunity to improve our paper.
We have edited the paper based on suggestions made by the reviewer; we reported used track changes for typos corrections in the manuscript and underlined in yellow the added paragraphs.
More in the details, following Reviewer  suggestions:

- we cited the current incidence of renal renal vein thrombosis and we tried to give more explanation and informations regarding parvus tardus and we eliminated the repetitions

I hope we were able to improve even more and to reach the value needed for publicaiton

In the meantime, we send to you my warmest regards, looking forward to reciving your feedback.

Your sincerely

Vito Cantisani and Emanuele David
